# Non-Secretor Status Due to *FUT2* Stop Mutation Is Associated with Reduced Rotavirus Infections but Not with Other Enteric Pathogens in Rwandan Children

**DOI:** 10.3390/microorganisms13051071

**Published:** 2025-05-03

**Authors:** Jean Bosco Munyemana, Jean Claude Kabayiza, Eric Seruyange, Staffan Nilsson, Gustaf E. Rydell, Anna Martner, Maria E. Andersson, Magnus Lindh

**Affiliations:** 1Department of Microbiology and Parasitology, School of Medicine and Pharmacy, University of Rwanda, Kigali P.O. Box 3286, Rwanda; munyebos1@gmail.com; 2Department of Infectious Diseases, Institute of Biomedicine, University of Gothenburg, 405 30 Gothenburg, Sweden; gustaf.rydell@gu.se (G.E.R.); maria.andersson.3@gu.se (M.E.A.); 3Department of Pathology, University Teaching Hospital of Kigali, Kigali P.O. Box 655, Rwanda; 4Department of Pediatrics, School of Medicine and Pharmacy, University of Rwanda, Kigali P.O. Box 3286, Rwanda; jckaba@gmail.com; 5Department of Pediatrics, University Teaching Hospital of Kigali, Kigali P.O. Box 655, Rwanda; 6Department of Internal Medicine, Rwanda Military Referral and Teaching Hospital, Kigali P.O. Box 3377, Rwanda; ericseruyange@gmail.com; 7Department of Internal Medicine, School of Medicine and Pharmacy, University of Rwanda, Kigali P.O. Box 3286, Rwanda; 8Department of Laboratory Medicine, Institute of Biomedicine, University of Gothenburg, 405 30 Gothenburg, Sweden; staffan.nilsson@gu.se; 9Department of Microbiology and Immunology, Institute of Biomedicine, University of Gothenburg, 405 30 Gothenburg, Sweden; anna.martner@gu.se

**Keywords:** *FUT2*, *rs601338*, secretor status, diarrhea, rotavirus

## Abstract

Enteric pathogens remain a health threat for children in low-income countries. A single nucleotide polymorphism (SNP) in the *FUT2* gene that precludes the expression of fucosyltransferase 2 has been reported to influence the susceptibility to rotavirus and norovirus infections. The aim of this study was to investigate the association between G428A at *rs601338* (stop codon variant) in the *FUT2* gene and a range of enteric pathogens in children under 5 years of age. Rectal swab samples from 668 children (median age 13.6 months, 51% males, 93% rotavirus vaccinated, 468 with diarrhea) from Rwanda were analyzed via PCR for pathogen detection and SNP genotyping. A *FUT2* stop codon (‘non-secretor’ status) was found in 19% of all children. Rotavirus was detected in 5.3% of non-secretors compared with in 13% of secretors (OR = 0.39, *p* = 0.019). Rotavirus P[8] was the predominant genotype and was found in 2.3% of non-secretors compared with 8.8% of secretors (*p* = 0.009). There was no association with any other pathogen, including noroviruses, of which 2 of 14 GII.4 infections were detected among non-secretors. Thus, the *FUT2* stop codon variant was associated with rotavirus but not with any other pathogen.

## 1. Introduction

Enteric infections pose a significant threat to the health of children under five in low- and middle-income countries [1] where living conditions and lack of hygiene strongly contribute to the transmission of enteric pathogens [2,3]. Host genetic factors, represented by single nucleotide polymorphisms (SNPs), may influence an individual’s susceptibility to and clearance of infections, for example, hepatitis C, HIV, malaria and norovirus [4,5].

The *FUT2* gene encodes a fucosyltransferase that is responsible for the presence of the so-called H antigen in secretions and mucosal tissues, including the gastrointestinal tract. People who are homozygotes for the A allele at the SNP *rs601338* (known as the G428A stop codon variant) have a non-functional *FUT2* gene and are referred to as non-secretors. They constitute approximately 20% of Caucasian and African populations and reportedly have a lower susceptibility for certain rota- and norovirus subtypes, such as the P[8] genotype of rotavirus and the GII.4 genotype of norovirus [6,7,8,9,10,11,12]. Both the circulating viral genotypes and *FUT2* genotypes may vary geographically, and studies from additional regions are warranted to better understand the impact of secretor status on the susceptibility to rotavirus and norovirus infections. It is also of interest to investigate the possible relation between secretor status and other pathogens than those already studied. Therefore, we have investigated the potential impact of *FUT2*-related secretor status on a broad range of enteric infections in among children in Rwanda, where such data are lacking.

## 2. Materials and Methods

### 2.1. Study Design, Study Area and Characteristics of Participants

Rectal swab samples from children who had participated in two recent studies of enteric infections in Rwanda were analyzed by *rs601338* SNP genotyping. One of them was cross-sectional and included 794 children with or without diarrhea between 6 September and 5 November 2021 [13]. The other was a study of 120 children who had a long-term follow-up after an episode of diarrhea and was conducted from 12 June to 15 December 2022. For the present study, we included the 668 children whose rectal swab samples contained enough human DNA to be genotyped for the *rs601338* SNP. These included, from the first study, 362 samples from children with diarrhea and 198 without diarrhea, and from the second study, 108 samples collected at the enrollment when the children presented with diarrhea. Only one sample per child was included.

The median age of study participants was 13.6 months (age distribution is shown in Figure 1), 51% were boys and 93% were rotavirus vaccinated. Seventy percent had diarrhea at the time of sample collection.

### 2.2. PCR for Pathogen Detection

Real-time PCR targeting a broad range of pathogens was performed as previously described [13], with primers and probes described in Appendix A. Briefly, 250 µL of rectal swab sample were mixed with 2 mL of lysis buffer followed by total nucleic acid extraction in an EasyMag instrument (Biomerieux, Marcy l’Étoile, France). The nucleic acids were eluted in 110 μL volume, and 5 μL of this was used for 45 cycles of PCR in a QuantStudio 6 flex instrument (Applied Biosystems, Foster City, CA, USA) in 8 parallel amplifications using a two-step setting (15 s at 95 °C, 60 s at 56 °C).

A real-time PCR specific for norovirus GII.4 was applied on samples positive for norovirus GII in the multiplex PCR (forward primer, GATGGGTCCACAGCCAACCT; reverse primer, CCGCTACAGGTGCCGCAA; probe ACGGGCTCCAAAGCCATAACCTCATT) [14]. Rotavirus genotyping was performed by real-time PCR as previously described [15].

### 2.3. FUT2 SNP Genotyping

For *FUT2* SNP genotyping at *rs601338*, a segment of the *FUT2* gene was amplified by 45 cycles of PCR (15 s at 95 °C, 60 s at 60 °C) with primers *FUT2*F, GCAGAACTACCACCTGAACGACT, *FUT2*R, GTGGTCGTGCAGGGTGAAC and allele-specific probes *FUT2*-FAM-MGB_G, CTGCTCCTGGACCTT and *FUT2*-VIC-MGB_A, CTGCTCCTAGACCTT in a Taqman genotyping master mix (Applied Biosystems). The genotype (AA, GA or GG) was identified by the allele analysis setting in the QuantStudio 6 Pro instrument (Applied Biosystems).

### 2.4. Data Analysis

The IBM SPSS Statistics 28.0 software (IBM Corporation, New York, NY, USA) was used for data analyses. The relationship between *FUT2* genotypes and pathogens was examined using a two-tailed Fisher exact test. Multiple logistic regression analysis was used with diarrhea as the outcome (dependent variable) and each pathogen and *FUT2* status as independent variables.

### 2.5. Ethical Approval

The study was approved by the University of Rwanda’s ethics committee (226/CMHS IRB/2021) and the University Teaching Hospital of Kigali’s ethics council (EC/CHUK/110/2021). Informed consent to participate was obtained from the children’s parents or legal guardians.

## 3. Results

### 3.1. Pathogen Association with Diarrhea and FUT2 Genotype

Among all study participants, 19% were homozygous for the *FUT2 rs601338* A allele and were thus defined as a non-secretor or *FUT2^−^*, while 42% had a GA genotype and 39% a GG genotype and thus were defined as *FUT2^+^*. The proportion of *FUT2*^−^ did not differ significantly between children with or without diarrhea (18% vs. 23%, *p* = 0.13). As shown in Table 1, only rotavirus was associated with the *FUT2* genotype, with a significantly (*p* = 0.019) lower frequency among non-secretors (7/132, 5.3%) than among secretors (67/536, 13%). Since the impact of secretor status might be confined to symptomatic infection, we compared the pathogen detection rates for children with or without diarrhea in secretors and non-secretors. Several pathogens were associated with diarrhea with relatively high odds ratios, including rotavirus, *Shigella*, ETEC, *Cryptosporidium* and *Campylobacter*, and as shown in Table 2 this association was apparent for children with the *FUT2*^+^ genotype (secretors). If non-secretors had been protected from diarrhea, then the OR would be lower than for secretors. This was seen only for rotavirus (OR 1.35 vs. 3.84), whereas other pathogens associated with diarrhea had a similar elevation of OR in secretors and non-secretors (but with *p* above 0.05 in the *FUT2*^−^ group, likely reflecting the low number of cases).

Multiple regression analysis showed that the association between the *FUT2*^−^ genotype and rotavirus remained significant when age, sex, and rotavirus vaccination were considered. ETEC-*eltB* was strongly associated with symptomatic infection with an OR = 3.67 (*p* < 0.0001) for secretors and had a lower OR (1.95, *p* = 0.21) for non-secretors, suggesting that non-secretors might be protected against diarrhea, but the difference was not statistically significant by interaction term analysis. A complementary analysis of the potential impact on symptoms is shown in Table 3, comparing the risk of diarrhea by *FUT2* status for children who were infected. Elevated odds ratios were seen for rotavirus, ETEC-*eltB* and *Salmonella*, suggesting that *FUT2*^−^ children might have a lower risk of symptomatic infection by these pathogens, but the associations were not statistically significant (*p* values 0.10–0.20).

### 3.2. Rotavirus P Types and Secretor Status

Samples positive for rotavirus were further genotyped for virus variants. Rotavirus P[8] was the most prevalent P genotype and was found in 50 of the 61 (82%) rotavirus-positive samples that could be genotyped. The number of cases was too low to accurately analyze differences between the P genotypes. As shown in Table 4, the P[8] genotype of rotavirus was detected in 2.3% (3/132) of *FUT2*^−^ compared with 8.8% (47/536) of *FUT2*^+^ children (*p* = 0.009), supporting that at least infection by this subtype is restricted by secretor status.

### 3.3. Norovirus GII and Secretor Status

Norovirus GII was detected in 52 of 536 secretors (9.7%) and in 17 of 132 non-secretors (13%), which contrasts with previously reported associations between non-secretor status and lower susceptibility to norovirus infection [6]. Since GII.4 has been the predominant genotype of norovirus GII in recent decades [16,17], we investigated the prevalence of this subtype with a specific real-time PCR [14]. This analysis identified GII.4 among 2/17 non-secretors (12%), compared with 12/52 secretors (23%, *p* = 0.49) with norovirus GII infection. The relatively small total number of GII.4 infections might, to some extent, be an underestimation due to the additional freeze–thawing prior to subtyping or to a lower sensitivity of the subtyping than with the initial PCR.

## 4. Discussion

The observed frequencies of *FUT2 rs601338* SNP genotypes in this study in Rwanda, with 19% carrying the AA genotype (stop codon homozygote, ‘non-secretor’), are similar to previous findings in Burkina Faso and South Africa, where 21% and 17% were non-secretors [6,18]. Rotavirus infections were significantly less common among non-secretors (OR = 0.39, *p* = 0.019) also when age, sex and rotavirus vaccination were considered, which aligns with previous studies suggesting that *FUT2* secretor status influences the susceptibility to rotavirus infection [9,12,19] by controlling the synthesis of glycans recognized by the virus [20]. In the first published studies, the non-secretor phenotype appeared to confer complete resistance against P[8] strains of rotavirus [10,19,21], but results from subsequent studies suggest that P[8] can also infect non-secretors [7]. We found that the P[8] genotype was less frequent in non-secretors than in secretors (2.3% vs. 8.8%, *p* = 0.009), supporting that non-secretors are significantly but not completely protected against P[8] infection. Likewise, a study from Tunisia found that 15% of the P[8]-infected subjects were non-secretors and proposed that this might reflect that preexisting conditions may influence susceptibility [22]. A more recent study from Tunisia, however, showed that the susceptibility was linked to the rotavirus P[8] subtype and that subtype P[8]-4 preferentially infects non-secretors while other P[8] strains, as well as P[4], preferentially infect secretors [23]. We found only one non-secretor among the 11 children infected with P[4], P[6] or P[9] rotavirus strains, suggesting possible protection (OR = 0.40), but the low number of patients precludes statistical evaluation (*p* = 0.70).

The lower rate of rotavirus infections among non-secretors that we observed reflects the situation among vaccinated children, who constituted 93% of the children in our study. Non-secretors might develop a poorer response to the vaccine [7], and it would be interesting to compare the impact of *FUT2* status in vaccinated and unvaccinated children, but the latter group was too small to allow this analysis.

Norovirus has also been shown to bind to *FUT2*-dependent glycans in a strain-dependent manner [24,25]. Non-secretors are protected from infection by strains from GII.4 [26] and possibly also GII.3 strains [27], while other less common strains such as GII.2 [28] seem to infect in a *FUT2*-independent manner [29]. Our finding of norovirus GII in 13% of non-secretors compared with 9.7% of secretors does not support that non-secretor status provides protection against the norovirus GII strains that circulate in Rwanda. This observation was surprising and suggests that secretor-independent strains of norovirus GII were more common than secretor-dependent strains in Rwanda at the time of our study. This possibility was supported by our use of an additional GII.4-specific PCR, which detected GII.4 in only 14 of the 71 norovirus GII-positive samples. Two of these 14 GII.4 strains were found in non-secretors (12%), compared with 12 in secretors (23%). This difference was not statistically significant (*p* = 0.49) but suggests that non-secretors might be less susceptible to GII.4 infection compared with other GII types.

Previous studies have reported *FUT2* activity to be associated with inflammatory bowel disease (IBD) [30] and *Campylobacter* infection [31], and secretors from Bangladesh have been reported to be less likely to have symptomatic enteric infections by certain ETEC strains [32,33]. We found that the odds ratios for diarrhea among secretors versus non-secretors were elevated for rotavirus, ETEC-*eltB* and *Salmonella* (however with *p* values 0.10–0.20), suggesting non-secretors might be protected from symptomatic infection by these pathogens. Further studies, including a higher number of non-secretors, and more detailed characterization of the pathogens (including subtyping of rotavirus and norovirus) are needed to clarify if and how secretor status might be associated with infection and diarrhea.

A strength of our study is that it explores possible associations between *FUT2* inactivation and infection susceptibility for a broad range of enteric pathogens, but we did not find any previously unrecognized association. It should be pointed out though that we only investigated a marker for *FUT2* activity, and that susceptibility for other infections might rather depend on *FUT3* which determines the expression of the Lewis antigen on intestinal mucosa [10,24]. The relatively low number of norovirus infections is a limitation that precludes conclusions about the impact of *FUT2* status on norovirus infection, and due to the cross-sectional design of the study potential influence by *FUT2* status on infection duration or severity over time could not be investigated.

In summary, we found a significant association between *FUT2*^−^ status and a lower frequency of rotavirus infection and specifically with rotavirus genotype P[8], but no association with infection with norovirus or other pathogens.

## Figures and Tables

**Figure 1 microorganisms-13-01071-f001:**
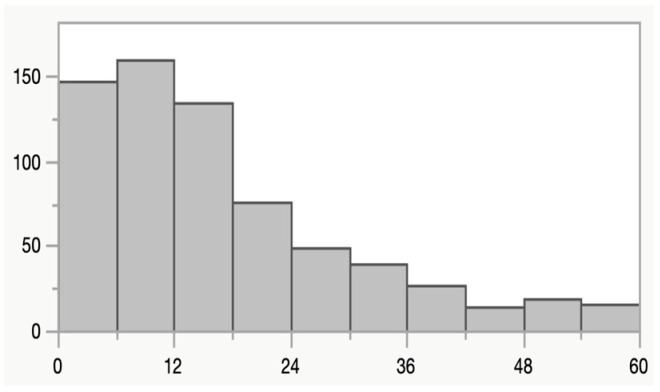
Age distribution (month of age on X axis) among the 668 children in the study.

**Table 1 microorganisms-13-01071-t001:** Pathogen detection rates by *FUT2* genotype.

	*FUT2*^+^ (n = 536)	*FUT2*^−^ (n = 132)	OR	*p*
Adenovirus 40/41 (n = 52)	7.1% (38)	11% (14)	0.64	0.20
Astrovirus (n = 30)	3.8% (25)	4.7% (5)	1.24	0.81
Norovirus GI (n = 24)	0.6% (21)	1.3% (3)	1.75	0.45
Norovirus GII (n = 69)	10% (52)	13% (17)	0.72	0.27
Rotavirus (n = 74)	13% (67)	5.3% (7)	2.55	0.019
Sapovirus (n = 78)	12% (62)	12% (16)	0.95	0.88
*Campylobacter* (n = 52)	7.8% (42)	7.6% (10)	1.04	1.0
ETEC-*eltB* (n = 158)	23% (125)	25% (33)	1.08	1.0
ETEC-*estA* (n = 81)	12% (67)	11% (14)	1.20	0.66
*Salmonella* (n = 38)	4.8% (26)	9.1% (12)	0.51	0.09
*Shigella* (n = 99)	15% (83)	12% (16)	1.32	0.41
*Cryptosporidium* (n = 22)	3.7% (20)	1.5% (2)	2.52	0.28

**Table 2 microorganisms-13-01071-t002:** Pathogen detection rates among children, stratified by *FUT2* genotype and diarrhea.

	*FUT2* ^+^				*FUT2* ^−^				Pint.
	Diarrhea	No Diarrhea	OR	*p*	Diarrhea	No Diarrhea	OR	*p*	
	(n = 384)	(n = 152)			(n = 86)	(n = 46)			
Adenovirus 40/41 (n = 52)	7.6% (29)	5.9% (9)	1.30	0.58	10% (9)	11% (5)	0.96	1.0	0.67
Astrovirus (n = 30)	5.2% (20)	3.3% (5)	1.62	0.50	4.6% (4)	2.2% (1)	2.20	0.66	0.80
Norovirus GI (n = 24)	4.2% (16)	3.3% (5)	1.28	0.81	2.3% (2)	2.2% (1)	1.07	1.0	0.90
Norovirus GII (n = 69)	11% (42)	6.6% (10)	1.74	0.15	14% (12)	11% (5)	1.33	0.79	0.69
Rotavirus (n = 74)	16% (60)	4.6% (7)	3.84	0.0003	5.8% (5)	4.4% (2)	1.35	1.0	0.30
Sapovirus (n = 78)	12% (45)	11% (17)	1.05	1.0	13% (11)	11% (5)	1.20	1.0	0.84
*Campylobacter* (n = 52)	9.6% (37)	3.3% (5)	3.13	0.012	10% (9)	2.2% (1)	5.25	0.16	0.65
ETEC-*eltB* (n = 158)	29% (110)	10% (15)	3.67	<0.0001	29% (25)	17% (8)	1.95	0.21	0.25
ETEC-*estA* (n = 81)	16% (67)	4.6% (7)	3.83	0.0003	14% (12)	4.4% (2)	3.57	0.14	0.94
*Salmonella* (n = 38)	5.7% (22)	2.6% (4)	2.25	0.18	8.1% (7)	11% (5)	0.73	0.75	0.17
*Shigella* (n = 99)	20% (75)	5.3% (8)	4.37	<0.0001	16% (14)	4.4% (2)	4.28	0.052	0.98
*Cryptosporidium* (n = 22)	4.9% (19)	0.7% (1)	7.86	0.02	2.3% (2)	0% (0)		0.54	0.62

Pint., *p* value for the potential impact by the interaction term (*FUT2* status) on diarrhea.

**Table 3 microorganisms-13-01071-t003:** Odds ratios for pathogen-associated diarrhea among *FUT2*^+^ and *FUT2*^−^ children.

	*FUT2* ^+^		*FUT2* ^−^			
	All	with Diarrhea	All	with Diarrhea	OR	*p*
Adenovirus 40/41 (n = 52)	38	29 (76%)	14	9 (64%)	1.79	0.48
Astrovirus (n = 30)	25	20 (80%)	5	4 (80%)	1.00	1.00
Norovirus GI (n = 24)	21	16 (76%)	3	2 (67%)	1.60	1.00
Norovirus GII (n = 69)	52	42 (81%	17	12 (71%)	1.75	0.50
Rotavirus (n = 74)	67	60 (90%)	7	5 (71%)	3.43	0.20
Sapovirus (n = 78)	62	45 (73%)	16	11 (69%)	1.20	0.76
*Campylobacter* (n = 52)	42	37 (88%)	10	9 (90%)	0.82	1.00
ETEC-*eltB* (n = 158)	125	110 (88%)	33	25 (76%)	2.35	0.10
ETEC-*estA* (n = 81)	67	60 (90%)	14	12 (86%)	1.43	0.65
*Salmonella* (n = 38)	26	22 (85%)	12	7 (58%)	3.93	0.11
*Shigella* (n = 99)	83	75 (90%)	16	14 (88%)	1.34	0.66
*Cryptosporidium* (n = 22)	20	19 (95%)	2	2 (100%)	0.00	1.00

**Table 4 microorganisms-13-01071-t004:** Presence of diarrhea among children with rotavirus P types stratified by *FUT2* genotype.

		*FUT2*^+^ (n = 536)		*FUT2*^−^ (n = 132)	
	n	Diarrhea	No Diarrhea	Diarrhea	No Diarrhea
Rotavirus	74	60	7	5	2
P[8]	50	42	5	2	1
P[4]	7	7	0	0	0
P[6]	3	2	0	1	0
P[9]	1	1	0	0	0

## Data Availability

The original contributions presented in this study are included in the article/Appendix A. Further inquiries can be directed to the corresponding author.

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
