# Peer review of "Non-Secretor Status Due to FUT2 Stop Mutation Is Associated with Reduced Rotavirus Infections but Not with Other Enteric Pathogens in Rwandan Children"

_microorganisms, 2025, doi:10.3390/microorganisms13051071_

Round 1

Reviewer 1 Report

Comments and Suggestions for Authors

Comments and Suggestions for Authors

In the manuscript microorganisms-3568208, entitled “ Nonsecretor status due to FUT2 stop mutation is associated with reduced rotavirus infections but not with other enteric pathogens in Rwandan children”, the possible relation between secretor status and diarrhea is investigated. For this purpose, the authors analyzed a single SNP in the human FUT2 gene [G428A at rs601338 (stop codon variant)] and a range of enteric pathogens other than rotavirus in children under five years old. They found a significant association between secretor status and rotavirus infection, specifically with rotavirus of genotype P[8], but no association with infection with norovirus or other pathogens.

While the authors corroborate existing data on rotavirus infection and host secretor status, they also offer novel insights into the association between the FUT2 gene and infection by other enteric pathogens - a topic that remains poorly understood.

This reviewer has several comments that need to be addressed.

Major comments:

  1. Line 29. It is not understood which variables are related to the indicated comparison “ (5.3% vs. 13%, OR=0.39, P=0.019)”
  2. Line 30: “Rotavirus P8 was the predominant subtype”: The authors should find a way to integrate this result into the study context, i.e. the secretor status of the children studied.
  3. Lines 39-41: The sentence is too generic and adds little to the context. Could you be more explicit in indicating which SNPs particularly have that effect?
  4. Lines 48-51: Consider rephrasing the statement “Data from different geographic regions are, however, not concordant, possibly due to differences in host genetics or circulating viral subtypes or strains, and therefore studies from additional regions are needed”, which is poor, and not entirely convincing in its argumentation.
  5. Lines 58-61: The authors mention that two different sorts of sampling from children who had participated in two recent studies were used: one cross-sectional (between September 6 and November 5, 2021) and one longitudinal (from June 12 to December 15, 2022). References should be provided here, or the authors should describe the sample populations (e.g., inclusion criteria, age range, median age, gender). Data about the children included in the longitudinal sampling are missing. Did they have diarrhea or not?

Furthermore, exclusion criteria should also be indicated, as it has been previously suggested that preexisting health conditions could condition the development of a rotavirus infection independently of the secretor-nonsecretor status (Ayouni S, et al. Rotavirus P[8] Infections in Persons with Secretor and Nonsecretor Phenotypes, Tunisia. Emerg Infect Dis. 2015;21(11):2055-2058. doi:10.3201/eid2111.141901).

The authors should justify why two different data sources were used and confirm whether all participants for the cross-sectional sampling were assessed only once during the two months.

  1. Line 71: The authors are requested to clarify what target pathogens were included for the diagnostic test described in this study. Furthermore, they do not mention the extraction method used, which could be of interest, considering the type of sample analyzed.
  2. Line 78: The authors should clarify whether the primers used for FUT2 SNP genotyping at rs601338 were newly designed for this study or derived from prior published work
  3. Lines 94-96: The information about the age and gender of the children studied and Figure 1 (whose purpose is unclear) should be moved to the Materials and Methods section, as they describe the population studied.
  4. Lines 96-97: In addition to the percentages that the authors wish to show in the Results section, detailed data on the number of children studied, controls, diarrheal case counts, and clinical characteristics would enhance the robustness of the findings and facilitate a more precise interpretation of the results.
  5. Lines 103-106:” … suggesting that non-secretors might have a slightly lower risk for diarrhea in general”. The relevance of this finding should be discussed in the Discussion section rather than here. Moreover, the magnitude of the difference seems insufficient to robustly support the proposed conclusion.
  6. The authors should include the sample size examined in Table 1 to provide proper context for the presented data. The utility of Table 1 is substantially limited without a clear indication of diarrheal status.
  7. Table 2 is incomprehensible. It needs to be redesigned for clearer data visualization. Tables 1 and 2 appear to contain overlapping/relevant data that could be effectively merged to create a more coherent data presentation and avoid unnecessarily fragmenting the information.
  8. Table 3: It would be recommended to show in bold only relevant information, as statistically significant associations in case they exist, and indicate their significances as footnotes in the tables.
  9. Lines 121-132:”Multiple regression analysis showed that the association between the FUT2–genotype and rotavirus remained significant when age, sex, and rotavirus vaccination were considered”. This statement needs to be discussed in the Discussion section.
  10. Lines 156-157: The statement “FUT2 secretor status determines the susceptibility to rotavirus infection” should be softened (“determines” is too categorical), as some of the cited studies do not indicate it; instead, they suggest that secretor status may partially influence rotavirus infection. Others have reported controversial results, suggesting that different host HBGAs polymorphisms may influence the binding affinity of different viral genotypes of rotavirus, and secretor status may result in a higher susceptibility, but not definitive.
  11. Lines 161-162, line 170, and lines 176-178: Results should not be reported in the Discussion section (again)
  12. From line 137 onwards, the use of the rotavirus P genotype designation must be reviewed, including the tables. The term 'subtype P8' is not the most correct designation in this context. According to nomenclature conventions, the P genotype designations of rotavirus should enclose the number in square brackets [ ]. Thus, e.g., P[8] genotype, P[4], should be used.
  13. In the Methods section, the P genotyping rotavirus method is missing.
  14. Line 196: “we found a significant association between FUT2– status and rotavirus infection” . Is FUT2– a typo?

Minor comments:

  1. Line 48. Revise punctuation.
  2. Line 70. Indicate country of origin
  3. Lines 82 and 83: “Applied Biosystems”: Please, indicate the trademark city and country of origin.

Reviewer 2 Report

Comments and Suggestions for Authors

This manuscript by Munyemana et al. investigated the association between the FUT2 G428A SNP at rs601338 and enteric pathogens in Rwandan children. The authors examined the presence of multiple pathogens in a cohort of 668 children, 93% of whom were rotavirus-vaccinated using robust PCRs. Notably, it discovered a protective effect against rotavirus in non-secretors (19% of the cohort, OR=0.39, p=0.019), confirming the previously established link between FUT2-mediated glycans and rotavirus susceptibility in the post-vaccination setting. However, the findings of this study are significantly constrained by three key limitations. Firstly, the cohort's high vaccination rate introduces unresolved ambiguity regarding whether FUT2 status directly affects infection outcomes or indirectly interacts with vaccine efficacy, leaving a critical confounder unaddressed. Secondly, the low incidence of norovirus undermines the statistical robustness and generalizability of the findings. Lastly, the lack of longitudinal data limits the analysis to static snapshots, excluding FUT2's potential influence on infection duration or severity over time. In addition, I found some concerns that need to be addressed.

  1. PCR methods for pathogen detection did not describe the list of pathogens and primers and the reference is not a direct citation, which does not contain the details on the method.
  2. The description of Data analysis is not sufficient. More information should be included, such as, were one-tailed or two-tailed tests used in Fisher's exact test? Which variables were included in the model for logistic regression? Were any interaction terms considered? What significance level was used for hypothesis testing (e.g., p < 0.05)?
  3. Lines 130 to 105, “The proportion of FUT2– tended to be lower among children with diarrhea than among children without diarrhea (18% vs. 23%, age adjusted OR=0.73, P=0.13), suggesting that non-secretors might have a slightly lower risk for diarrhea in general.” is not accurate, statistically there is no difference.

Author Response

Comments and Suggestions for Authors

This manuscript by Munyemana et al. investigated the association between the FUT2 G428A SNP at rs601338 and enteric pathogens in Rwandan children. The authors examined the presence of multiple pathogens in a cohort of 668 children, 93% of whom were rotavirus-vaccinated using robust PCRs. Notably, it discovered a protective effect against rotavirus in non-secretors (19% of the cohort, OR=0.39, p=0.019), confirming the previously established link between FUT2-mediated glycans and rotavirus susceptibility in the post-vaccination setting. However, the findings of this study are significantly constrained by three key limitations.

Firstly, the cohort's high vaccination rate introduces unresolved ambiguity regarding whether FUT2 status directly affects infection outcomes or indirectly interacts with vaccine efficacy, leaving a critical confounder unaddressed.

RE: This is an interesting point. Non-secretors might bind the vaccine strain less well and develop a poorer response to the vaccine, as indicated by findings by Sharma et al (Viruses 2020). The protection against rotavirus from being non-secretor could therefore be different in unvaccinated than in vaccinated children. The number of unvaccinated children was too low thought to investigate this in our study. We have added a comment about this in Discussion.

Secondly, the low incidence of norovirus undermines the statistical robustness and generalizability of the findings.

RE: We agree and have added a comment about this limitation.

Lastly, the lack of longitudinal data limits the analysis to static snapshots, excluding FUT2's potential influence on infection duration or severity over time.

RE: We agree, but this is a limitation also with previous cross-sectional studies.

In addition, I found some concerns that need to be addressed.

  1. PCR methods for pathogen detection did not describe the list of pathogens and primers and the reference is not a direct citation, which does not contain the details on the method.

RE: A table showing the primers and probes has been added as supplementary material.

  1. The description of Data analysis is not sufficient. More information should be included, such as, were one-tailed or two-tailed tests used in Fisher's exact test? Which variables were included in the model for logistic regression? Were any interaction terms considered? What significance level was used for hypothesis testing (e.g., p < 0.05)?

RE: The Fisher’s exact test was two-tailed. The logistic regression analysis analyzed the outcome diarrhea (dependent variable) with one pathogen and FUT2 status as independent variables. Significance level was P<0.05. These clarifications have been added.

  1. Lines 130 to 105, “The proportion of FUT2– tended to be lower among children with diarrhea than among children without diarrhea (18% vs. 23%, age adjusted OR=0.73, P=0.13), suggesting that non-secretors might have a slightly lower risk for diarrhea in general.” is not accurate, statistically there is no difference.

RE: We have changed to “The proportion of FUT2 did not differ significantly between children with or without diarrhea (18% vs. 23%, P=0.13).”

Reviewer 3 Report

Comments and Suggestions for Authors

This is a report of Swedish-Rwandan collaborative study on etiology of childhood diarrhea in Rwandan children. More specifically, the present paper reports association of so-called non-secretor status with protection against certain diarrheas, notably rotavirus and norovirus. This topic has been subject of several Swedish studies for 20 years.

The present study consists of three components: 1. clinical material of stool samples from 668 hospitalized children (age distribution shown), mostly but not exclusively with diarrhea, 2. PCR studies for multiple enteric pathogens including rotavirus, norovirus GII and GI, sapovirus, astrovirus and enteric adenovirus. 3. Determination of the H-antigen non-secretor status by FUT2 SNP genotyping.

In short, the materials and methods of the study are in order, and the study has been carefully conducted. The results are as expected.

It was well known prior to this study that the risk of rotavirus diarrhea is lower in non-secretor children. In this study the finding is once again confirmed. Since most (93%) of the children were vaccinated with rotavirus vaccine the result applies to breakthrough infections of rotavirus. Interestingly, the rate of rotavirus infections (mostly diarrhea) seems to be about the same as the rate of norovirus or sapovirus diarrhea without vaccination.

The FUT2 (non-secretor) status was found to be significantly associated with protection against rotavirus, specifically P-type P[8]. Other associations were not statistically significant, although it seemed that there might be protection against other enteric pathogens as well. I do not believe that further studies with larger materials are warranted, but the present study might be regarded as final on this topic.

Author Response

We notice and appreciate that this reviewer had no comments requiring revision of our manuscript.

Round 2

Reviewer 1 Report

Comments and Suggestions for Authors

In response to the previous Reviewer's major comment, the authors noted that no exclusion criteria were applied. Thus, this reviewer suggests that the authors should address in the manuscript the limitation of not evaluating how pre-existing health conditions might affect rotavirus infection risk irrespective of secretor status. Otherwise, please indicate whether they did so.

Author Response

In response to the previous Reviewer's major comment, the authors noted that no exclusion criteria were applied. Thus, this reviewer suggests that the authors should address in the manuscript the limitation of not evaluating how pre-existing health conditions might affect rotavirus infection risk irrespective of secretor status. Otherwise, please indicate whether they did so.

RE: The study by Ayouni et al. found that 4 out of 30 patients with rotavirus P[8] were non-secretors (13%). Since previous studies had not observed P[8] in non-secretors they speculated in Discussion “P[8] infection of nonsecretors might be associated with preexisting health conditions, and healthy nonsecretors might never be infected by P[8] rotavirus.” A more recent study by Khachou et al. (JID 2020) has however shown that the likely explanation is not preexisting health conditions but rather the rotavirus subtype.

In the revised manuscript we cite and comment these articles in Discussion (line 176-180) to better describe the relation between rotavirus P[8] and secretor status. We mention that the study by Ayouni proposed preexisting conditions as an explanation and that the more likely explanation is subtype-dependence as shown in the study by Khachou et al.

Reviewer 2 Report

Comments and Suggestions for Authors

Thank you for revising your manuscript and addressing the issues raised during the previous review.

I appreciate the clarifications, especially on the study's limitations. Based on these revisions, I recommend that the manuscript be accepted for publication.

Author Response

Thank you for revising your manuscript and addressing the issues raised during the previous review.

I appreciate the clarifications, especially on the study's limitations. Based on these revisions, I recommend that the manuscript be accepted for publication.

RE: We notice that no further revision is required. Thanks.